# Sandwich-Based Immunosensor for Dual-Mode Detection of Pathogenic F17–Positive *Escherichia coli* Strains

**DOI:** 10.3390/ijms23116028

**Published:** 2022-05-27

**Authors:** Imed Salhi, Amal Rabti, Asma Dhehibi, Noureddine Raouafi

**Affiliations:** 1Livestock and Wildlife Laboratory (LR16IRA04), Arid Lands Institute (IRA), University of Gabès, Medenine 4117, Tunisia; salhi_imed@yahoo.fr (I.S.); asmadhehibi@gmail.com (A.D.); 2Analytical Chemistry and Electrochemistry Lab (LR99ES15), Sensors and Biosensors Group, Faculty of Science, University of Tunis El Manar, Tunis Manar 2092, Tunisia; amal.rabti@fst.utm.tn

**Keywords:** electrochemical, fluorescence, sandwich-based immunosensor, F17 fimbriae, *Escherichia coli*, pathogen

## Abstract

Bacterial diseases cause tremendous economic losses due to high morbidity and mortality in livestock animals. F17A protein, the major subunit of F17 fimbriae, is one of the most prevalent and crucial virulence factors among the pathogenic *Escherichia coli* (*E. coli*) isolated from diarrheic and septicemic animals of various species. Purification and detection of this protein is regarded as an interesting field of investigation due to its important role as a therapeutic target, such as vaccines, and as a diagnostic tool. In this context, polyclonal rabbit antibodies recognizing F17A protein (anti−F17A antibody) were developed and used for its detection. In fact, sandwich biosensor using anti−F17A/gold nanoparticles conjugates as capture probe and anti−F17A antibody labelled with horseradish peroxidase as signal amplification probe was developed for electrochemical and fluorescent detection of purified F17A protein and live F17–positive *E. coli* bacteria. Good specificity and sensitivity for detection of F17–positive *E. coli* strains were obtained. The dynamic range for the biosensor varies from 1 × 10^2^ to 1 × 10^9^ CFU·mL^−1^ (R^2^ = 0.998) and the detection limit (LOD) and the IC_50_ value were estimated to be 37 CFU·mL^−1^ and 75 CFU·mL^−1^, respectively.

## 1. Introduction

*E. coli* are part of the facultative aerobic-anaerobic gut microbiota of humans and animals. Commensal strains of *E. coli* play an essential role for the host by supporting nutrient delivery, maturation of the immune system, and protection of the gastrointestinal tract from colonization by other pathogenic microorganisms [1,2]. However, they can be associated with many intestinal and extra-intestinal pathologies through the acquisition of virulence factors (mainly fimbrial and afimbrial adhesins) [3,4].

F17 fimbriae are narrow rigid cylinders (3–4 nm in diameter), anchored to an outer membrane protein, F17C. Their structure is composed of two subunits: the major subunit F17A with four variants (F17Aa, F17Ab, F17Ac and F17Ad) and the minor subunit F17G with two variants F17G-1 and F17G-2 [5,6,7,8]. Recently, three other variants named F17Ae, F17Af and F17G-3 were identified in diarrheic cattle, suggesting their association with diarrhea in this species [6]. F17 fimbriae are produced mainly by *E. coli* strains isolated from ruminants, in association or not with clinical intestinal or septicemic disorders, especially in neonates, but sometimes also in adult animals [6]. In fact, we recently found that camels affected by diarrhea expressed a higher prevalence of fimbriae F17 than clinically healthy camels (46.5% vs. 14.5%) [9]. Several other studies also reported the association of this adhesin with diarrhea and/or sepsis observed in newborn alpacas [10], calves [11,12,13,14], lambs [5,15] and horses [16]. All this justifies the growing interest in this virulence factor as a therapeutic target and for the design of diagnostic tools such as ELISA and biosensing platforms. It is therefore essential to have the proteins that make up the pilus, including F17A in its pure state. Although pilin F17A was first purified directly from pathogenic strains [17] and F17A was produced as a recombinant protein from an isolated mastitis strain [18], cloning, expression, and purification of this protein are still needed.

General interest on biosensors using nanomaterials such as graphene [19], carbon nanotubes [20], gold nanoparticles [21,22], magnetic beads [23] and silicon nanowires [24] for protein detection and quantification applications is increasingly growing owing to their high sensitivity, specificity, and speed at lower costs. Andrei et al. developed a surface-enhanced Raman spectroscopy (SERS)-based biosensor for the detection of uropathogenic *E. coli* strains containing type-1 fimbriae in urine [25]. They used cetyltrimethyl ammonium bromide−stabilized gold nanorods, as SERS markers, and anti−fimbrial antibodies against the major pilin protein FimA as surface probes. Moreover, Rabti et al. designed an impedimetric multiplexed DNA biosensor based on gold nanoparticle−modified screen-printed carbon electrode for simultaneous detection of *E. coli* (yaiO gene) and its virulent F17–positive variant [21]. However, development of biosensors for the detection of variant fimbriae expressed by pathogenic *E. coli* is still a necessity.

Detecting the whole bacteria can be advantageous since it allows circumventing the DNA extraction, purification, and PCR amplification, which could be expensive and time consuming. Several recent works reported the use of sandwich-based approach for the detection of pathogenic bacteria. For instance, Bu et al. used a methylene blue loaded compositive as a signal-amplifying agent and magainin I as a bioreceptor to detect *E. coli* O157:H7 bacterial strain with a LOD of 32 CFU·mL^−1^ [26]. An interesting sandwich-based concept for the detection Shiga toxin-producing *E. coli* (STEC) was recently reported by Quintela and Wu [27]. Indeed, the authors used two bacteriophages, the first one was functionalized with biotin and served for surface tethering and capturing the bacterium and the second one was labelled with HRP for signal transduction. The biosensor is not selective for a particular strain, but rather it can detect several ones with a limit of detection varying from 10 to 100 CFU·mL^−1^ for STEC O157, O26, and O179 strains, respectively. A set of two aptamers was selected for *Staphylococcus aureus* and used to build an electrochemical biosensor. The limit of detection of 39 CFU·mL^−1^ was achieved for a spiked buffer solution and the LOD was tenfold higher in tap water [28]. A hybrid sandwich using one antibody and one aptamer was used to selectively detect *Vibrio parahaemolyticus*. The aptamer serves as starting point for rolling circle amplification in order to amplify the readout signal, to achieve a LOD as low as 2 CFU·mL^−1^ [29].

In this work, we report the expression and purification of recombinant F17A protein and the production of a rabbit polyclonal anti−F17A antibody. A sandwich immunosensor for the detection of recombinant and native F17A protein in F17 fimbriae-positive *E. coli* bacteria was generated by conjugating the anti−F17A antibody with gold nanoparticles and horseradish peroxidase (HRP), respectively. The use of the HRP, redox enzyme capable of catalyzing the oxidation of fluorogenic compounds into an electroactive and fluorescent substrate, allowed electrochemical and fluorescent detection of purified and native F17A protein and differentiation between F17 fimbriae positive and negative *E. coli* strains.

## 2. Results

### 2.1. Expression and Purification of F17A, Development and Validation of the Polyclonal Antibody

The pHAT-f17a plasmid DNA (Figure 1A) was transformed into competent cells of BL21 (DE3) after isolation. From the transformation plate, positive colonies were selected and approved by colony PCR. The construct was verified by sequencing three different clones. BLAST analysis against Uniprot database showed that the protein sequence was identical to the F17Aa isoform. Recombinant F17A protein was expressed and purified using SDS-PAGE as shown in Figure 1B. The molecular weight of the F17A protein was calculated to be around 20 kDa. The protein concentration was estimated by the OD at 280 nm considering the calculated extinction coefficient of the pure protein of 12,045 M^−1^cm^−1^ (calculated by ProtParam tool: https://web.expasy.org/protparam/ (accessed on 19 October 2020) and found to be 1.5 mg·mL^−1^. The purified recombinant F17A and native F17 from *E. coli* F17−positive strain lysates were detected with western blotting using the developed antibody.

Reactivity with whole bacterial cells showed that the antibody can recognize the F17A in its native form on the cell surface that is essential for the following steps. This result is confirmed by the recognition of the antigen by Western blot with the purified protein and a cell lysate of *E. coli* F17−positive bacteria (Figure 1B). Note that only a single band is detected for cell lysate showing that the antibody does not cross-react with other bacterial proteins.

To assess the specificity and capability of the prepared polyclonal rabbit anti−F17A antibody, indirect protein, and whole-cell ELISA (enzyme linked immunosorbent assay) experiment were performed. Different concentrations of the purified antibody were used on F17A protein (5 µg·mL^−1^) or *E. coli* F17–positive strain (10^6^ CFU·mL^−1^) coated ELISA plates. For the purified protein ELISA, we used the purified antibody concentrations ranging from 40 to 2000 ng·mL^−1^ with detection limit around 100 ng·mL^−1^ (Figure 2A). For the cell−based ELISA, antibody concentration ranged from 100 to 2000 ng·mL^−1^ with detection limit around 200 ng·mL^−1^ (Figure 2B).

### 2.2. Building the Biosensor

To detect F17A protein and isolated F17–positive and -negative *E. coli* strains, a sandwich-based electrochemical biosensor was developed using polyclonal antibody anti−F17/AuNPs (gold nanoparticles) conjugate electrochemically deposited onto screen-printed carbon electrode (SPCE) and a second antibody anti−F17A labeled with HRP as signal amplification (Figure 3A). The electrocatalytic reduction of HQ as redox mediator in the presence of H_2_O_2_ as enzyme substrate served for the analytical differential pulse voltammetry (DPV) and chronoamperometry readouts.

The conception of the fluorescence-based bioassay comprises the following steps of (1) connecting the polyclonal rabbit anti−F17A antibody onto the surface of gold nanoparticles via physical bonding; (2) capturing the target F17A protein or *E. coli* F17–positive strains with the capture antibody; (3) adding the revealing horseradish peroxidase (HRP) conjugated antibody (HRP-Ab); (4) incubation for a few minutes with a solution containing o-phenylenediamine and hydrogen peroxide; and (5) adding shortly after (15 min) cyanide ions to block the enzymatic activity (Figure 3B).

### 2.3. Electrochemical Immunosensing

#### 2.3.1. EIS Characterization

EIS was applied to monitor each modification step of the SPCE using the redox ferricyanide/ferrocyanide couple (Figure 4A). The Nyquist plots demonstrated that the charge-transfer resistance of Ab−AuNPs/SPCE (R_ct_ = 571.3 ± 2.9 Ω (*n* = 3)) is lower than that of bare SPCE (R_ct_ = 787.5 ± 2.4 Ω), due to the enhancement of the electro−active surface area and conductivity of the working electrode by the Ab−AuNPs conjugate. After incubation in F17A protein solution, the resistance further decreased (R_ct_ = 469.1 ± 3.6 Ω), which is probably related to the attractive ionic interactions between the positively charged F17A protein layers and the negatively charged redox ions [21]. Subsequently, the R_ct_ value increased (R_ct_ = 591.5 ± 3.2 Ω) after incubation of HRP−Ab as the sandwich complex formation impede the diffusion of the redox probe toward the electrode surface to be oxidized or reduced. These results confirmed the effective formation of the biosensor.

#### 2.3.2. Optimization Steps

Since the outcome response is dependent on several experimental parameters such as the concentration of antibody, the time of immobilization of polyclonal antibody anti−F17 onto AuNPs surfaces, the time of immobilization of *E. coli* F17–positive onto Ab−AuNPs/SPCE, the time of immobilization of Ab−HRP onto F17/Ab−AuNPs/SPCE, and the Ab−HRP concentration, we optimized them to achieve higher performances. The ΔI value, representing the difference between the current measured in the absence and presence of *E. coli* F17–positive strains, was used to evaluate the biosensor sensitivity.

As shown in Figure 4B,C, the maximum ΔI was obtained when the antibody concentration is 25 µg·mL^−1^ and the time of its immobilization onto AuNPs surfaces is 60 min. Regarding the incubation time of F17A protein, the ΔI increased until 30 min, while longer incubation time did not strongly affect the response, and thus 30 min was chosen as the incubation time in subsequent experiments (Figure 4D). Moreover, the results in Figure 4E demonstrated that 15 min was enough to ensure optimal conjugation with HRP−labeled antibody. Furthermore, the influence of the dilution ratio of Ab−HRP was also assayed. As displayed in Figure 4F, ΔI value decreased when the dilution ratio changed from 1/60 to 1/500 while further increasing the dilution ratio resulted in poor conjugation with *E. coli* (F17–positive)/Ab−AuNPs/SPCE. As no significant difference was obtained when using 1/60 and 1/125 dilution ratio, the dilution ratio of Ab−HRP of 1/125 was used in the following assays.

#### 2.3.3. Analytical Characteristics of the Electrochemical Bioassay

As shown in Figure 5A, only an anodic peak at ca. +0.125 V vs. Ag/AgCl related to the oxidation of hydroquinone (HQ) was observed in the DPV response of the developed immunosensor in absence of the target biomolecules (i.e., F17A, *E. coli* F17+ or *E. coli* F17−) (curve a). When F17A protein or *E. coli* F17–positive strain was added, a cathodic peak at −0.140 V is observed, corresponding to the electrochemical reduction of benzoquinone (BQ), resulting from the oxidation product of HQ by HRP in presence of H_2_O_2_, while the anodic peak current was significantly reduced (curves b and c). This may be explained by the successful formation of the sandwich biosensor and, respectively, the catalytic oxidation of HQ by the HRP conjugated antibody (HRP−Ab) according to the following equations:HRP(red)−Ab + H_2_O_2_ → HRP(ox)−Ab + H_2_O(1)
HRP(ox)−Ab + HQ → HRP(red)−Ab + BQ(2)
where, HRP(red) and HRP(ox) are, respectively, the reduced and oxidized forms of the heme catalytic center of the HRP.

However, in the presence of *E. coli* F17−negative strain, small cathodic and anodic peaks were observed (curve d), indicating a reduced oxidation of HQ molecules and respectively the small amount of adsorbed HR−Ab on the electrode surfaces. This can be due a small nonspecific recognition of *E. coli* F17−negative strain by the developed biosensor, thus demonstrating its good selectivity.

Even known that direct detection of F17A protein in its native form, i.e., *E. coli* bacteria positive for F17 fimbriae, is better for practical application, determination of different concentrations of *E. coli* F17–positive was carried out. In fact, the intensity of the cathodic peak in the DPV measurements related to the electrochemical reduction of BQ was significantly enhanced with increasing concentrations of *E. coli* F17–positive in the range from 1 × 10^2^ to 1 × 10^9^ CFU·mL^−1^. A generated calibration curve was then plotted in Figure 5B with log scale of concentration (x-axis). Knowing that logistic model is recommended for immunoassays involving enzyme-labeled antibodies and labeled−antibody, [30] the data points were fitted to a non−linear 4-parameter logistic (4-PL) model using OriginPro 8 software as follow:Y = A2 + (A1 − A2)/(1 + (x/x0)^p)(3)
where, A1 and A2 are, respectively, the responses at high and low asymptote; x_0_ is the concentration corresponding to 50% of specific binding (IC_50_); and p is the slope factor.

The obtained adjusted R-squared (R^2^) of the experimental data was 0.998 for a working range of 1 × 10^2^ – 1×10^9^ CFU·mL^−1^. The limit of detection (LOD) calculated using the formula “blank mean +3.3×the standard deviation of blank”, was estimated to be 37 CFU·mL^−1^ and the IC_50_ value was 75 CFU·mL^−1^. The calculated limit of quantification was 112 CFU·mL^−1^.

The reproducibility and stability of the prepared immunosensor were evaluated from the results of amperometric detection of *E. coli* F17–positive (10^6^ CFU·mL^−1^) using three independently prepared biosensors or by recording amperometric response over four weeks storage at 4 °C. The RSD of the peak current was ca. 6% revealing an acceptable reproducibility of our electrode. Moreover, the biosensor retained about 89% of its initial current, which confirms a good stability.

### 2.4. Fluorescence Bioassay

To verify the viability of this strategy, comparison of the obtained responses with and without F17A protein was examined. As depicted in Figure 5D, a relatively low fluorescence signal was observed in the absence of F17A protein, probably related to the modified-gold nanoparticles catalyzed oxidation of o-phenylenediamine (OPD) by H_2_O_2_. Conversely, in the presence of F17A protein, fluorescence emission of the biosensor was much improved under the same conditions, which is due to the presence of HRP−Ab attached on the conjugated complex of F17A protein/Ab − AuNPs. The specificity of the developed immunosensor was further evaluated by testing the response of the F17 positive and negative *E. coli* strains. As it turns out, the fluorescence signal in the presence of *E. coli* F17–positive strains was clearly distinguished from *E. coli* F17− strains (Figure 5D), which proved the high selectivity of this assay owing to the highly specific recognition between F17A and its antibodies.

## 3. Discussion

Biosensors have been one of the most attractive detection techniques due to their excellent performance, namely cost-effectiveness, fast response, high sensitivity, and high selectivity [31,32,33,34,35,36,37]. Immunosensors are a very attractive tool for the detection of pathogenic bacteria. They are based on the use of poly or monoclonal antibodies, which recognize a target on the surface of bacterial cells. This target must be unique to specifically identify the pathogenic strain. In a previous work, we showed that F17 fimbria was associated with neonatal diarrhea in camel calves [9]. In this work, we cloned, expressed, and purified the major subunit of this fimbria, F17A. Since the His-tag is N-terminal, the protein was expressed as inclusion bodies [38] and we had to solubilize it by urea (8.0 M) and refold it by buffer exchange against PBS. Our results are similar to those found by Chen et al. [18]. The purified and desalted protein tended to aggregate rapidly and precipitate which may be explained by the fact that protein A constitutes the monomer of the fimbriae filament.

We developed a rabbit polyclonal anti−F17A antibody that specifically recognized the F17A protein, in its recombinant or native forms on live F17–positive *E. coli* bacteria, using indirect protein and whole cell ELISA experiments and western blotting analyses. Taking advantage of the sandwich approach, using the nanoscale properties and biocompatibility of AuNPs, which increase the capture antibody loading, and HRP activity to catalyze the oxidation of a fluorogenic molecule intro an electroactive and fluorescent product, we designed an immunosensor for electrochemical and fluorescent detection of purified and native F17A protein with increased specificity and sensitivity. The developed test has a wide range of applications and can be used to detect other pathogenic bacteria and viruses. Although the proof-of-concept of the fluorescence-based biosensor has been validated, detailed studies of the biosensor parameters are limited since the centrifugation and ultrasound treatments required for separation and redispersion of AuNPs conjugates can eradicate *E. coli* bacteria.

Although no electrochemical biosensor was reported for detection of F17–positive *E. coli* bacteria, our biosensor showed satisfactory analytical performances compared to other biosensors of pathogenic *E. coli* (Table 1). For instance, the developed immunosensor showed better LOD than that of *E. coli* O157:H7 impedimetric biosensor (LOD = 48 CFU·mL^−1^), based on the use of self-assembled gold nanoparticles and protein G [39], or *E.*
*coli* O157:H7 biosensor combining aptamer−induced catalytic hairpin assembly circle amplification with microchip electrophoresis (LOD = 75 CFU·mL^−1^) [40]. Moreover, the elaboration of the sandwich biosensor and the detection of F17–positive *E. coli* had lower assay time. For instance, although sandwich−based electrochemical biosensor of *E. coli* O157:H7 had better LOD (LOD = 32 CFU·mL^−1^) [26], the aptamer immobilization as the capture probe onto the electrode surfaces required an overnight incubation time, whereas in our case only ~2.5 h were enough for Ab−AuNPs preparation and electrodeposition. Moreover, 30 min was sufficient for *E. coli* immobilization onto our modified electrode surfaces while Bu et al. needed 2 h for *E. coli* O157:H7 immobilization step. Hence, the designed biosensor achieved excellent performance for F17–positive *E. coli* which may be credited to the high surface area of gold nanoparticles allowing better loading of antibodies and to the strong interaction between the developed polyclonal rabbit anti−F17A antibody and F17–positive *E. coli* strains.

As displayed in Figure 5C, a lower amperometric signal is obtained in the presence of F17A protein, which agrees with the successful formation of the sandwich biosensor. Moreover, the ΔI was calculated to be –6.74 µA for *E. coli* strains F17–positive sample and –2.84 µA for the F17-negative sample (assuming 100% recognition for F17–positive sample, almost 41% of the response was obtained in the presence of F17-negative sample), which indicated the good discrimination between both samples.

## 4. Materials and Methods

### 4.1. Cloning, Expression and Purification of F17A

An *E. coli* strain of virotype: f17/afa/EastI/papC/iroN/iss/iucD, serogroup O64 and belonging to phylogenetic group B1 isolated from fecal samples of a diarrheic camel calf [9], was used for the extraction of the nucleotide sequence encoding the F17A antigen. Based on the f17a gene sequence, a set of primers was designed; f17a-for: 5′-TATTATCCATGGTATGACGGTACAATTACTTT-3′, containing an *Nco*I site (underlined) and f17a-rev: 5′-TATTGCGGCCGCTTACTGATAAGCGATGGTGT-3′, containing a *Not*I site (underlined). The PCR steps were as follows: initial denaturation at 95 °C for 5 min, then 30 cycles at 95 °C for 45 s and annealing at 55 °C for 45 s followed by application of 72 °C for 60 s, and final extension for 10 min at 72 °C. The resulting PCR amplified f17a gene was cloned into the pHAT vector (kindly provided by Dr Marko Hyvonen, Cambridge University, UK) after cleavage with the two restriction endonucleases (Promega, Madison, WI, USA). The obtained sequence was then transformed into competent *E. coli* BL21 (DE3) bacteria. The bacteria were grown for 4 h at 37 °C at 250 rpm in 500 mL of LB medium and the expression of the F17A protein was induced by 1 mM IPTG (isopropyl-D-thiogalactopyranoside; Thermo Scientific, Waltham, MA, USA) for 4 h. Lysis buffer (8 M urea, 500 mM sodium chloride, 10 mM Tris, pH 8.0) was used to harvest and resuspend the cells before being centrifuged for 30 min at 15,000× g. the recombinant protein was purified by fast protein liquid chromatography (FPLC) on a Hitrap IMAC column (Cytiva, Uppsala, Sweden) with an equilibration buffer (phosphate buffer 20 mM pH 7.4, 500 mM NaCl, 6 M urea and 20 mM imidazole) and an elution buffer (phosphate buffer 20 mM pH 7.4, 500 mM NaCl, 6 M urea and 500 mM imidazole) then renatured by passing through a Hitrap Desalting column (Cytiva, Uppsala, Sweden) against PBS buffer at pH 7.4. SDS-PAGE was used to assess the protein purity (Figure 1).

### 4.2. Development of Polyclonal Antibodies against F17A

All experiments were performed in compliance with the policies of the first author’s institute on animal use and ethics. Polyclonal antibodies were produced in one New Zealand rabbit using the purifiedF17A protein. A first injection of a mixture of 1 mL F17A (0.3 mg in PBS) and 1 mL of complete Freund adjuvant was injected subcutaneously to the rabbit, followed by four boosts with incomplete Freund adjuvant. Three days after the last boost, serum was collected through centrifugation of the coagulated blood of the studied rabbit. FPLC was used to purify the antibodies from rabbit serum. After equilibration of the column with 15 mL of BP, the rabbit IgGs were eluted with 0.1 M citrate buffer pH 3, quantified at 280 nm and stored at −30 °C. 

### 4.3. Enzyme Linked Immunosorbent Assay and Western Blot

Multiscan Sky ELISA plate reader (Thermo scientific, Waltham, MA, USA) was used to perform all ELISA assays. First, 96−well microplates (Greiner Bio-One, Kremsmünster, Austria) were coated overnight at 4 °C with 100 μL per well of 5 µg·mL^−1^ solution of the purified protein in PBS or of about 106 cells/well bacterial suspension obtained after culture of F17–positive *E. coli* strain in LB medium overnight at 37 °C, centrifugation at 4000× *g* for 15 min and re-dispersion in PBS. The microplates were emptied, dried with an air dryer. Then, 200 μL of 3% BSA in PBS were used to block the residual binding sites for two hours. The rabbit anti−F17A antibody was used at various concentrations (100 µL from 40 to 2000 ng·mL^−1^ for the purified F17A and from 100 to 2000 µg·mL^−1^ for F17–positive strain). Anti−Rabbit HRP conjugated antibody (Invitrogen, Waltham, MA, USA) (100 μL from 1/5000 dilution) served as a secondary antibody and 100 μL of OPD as a substrate. After 20 min the reaction was stopped by 50 µL 3M HCl solution and optical density was read at 492 nm.

Expression of recombinant F17A and native protein in F17−positive *E. coli* strain was identified using western blot. Purified protein and F17–positive lysates were loaded in each well in SDS-PAGE. The developed Polyclonal rabbit anti−F17A antibody (1/2000 dilutions) and anti−rabbit HRP conjugated antibody (1/5000 dilutions) were applied, respectively, to probe the blots and as a secondary antibody. Blots were developed using SIGMAFAST™ 3,3′−diaminobenzidine western blot substrate (Sigma-Aldrich, Saint-Louis, MO, USA).

### 4.4. Preparation of AuNPs and Ab−AuNPs

Standard citrate reduction procedure was used to prepare AuNPs. Briefly, a mixture of 500 µL of 0.01 M chloroauric acid and 19.50 mL of deionized water was stirred and refluxed. Then 1 mL of 38.80 mM trisodium citrate solution was added, and the mixture kept boiling for 20 min until color changed to wine−red, suggesting the formation of gold nanoparticles [46,47,48,49,50]. Finally, the colloidal gold solution was cooled down to room temperature (RT), subsequently filtrated using 0.45-micron syringe and stored in the dark at 4 °C for further use.

Polyclonal rabbit anti−F17A antibody (Ab) was immobilized onto AuNPs surfaces by physical adsorption according to a literature procedure [51,52,53,54]. 25 µL of Ab (0.50 mg·mL^−1^) was added to 500 µL of the citrate capped AuNPs solution after adjusting the pH to 9.6 by adding 0.1 M Na_2_CO_3_ solution. Using a thermoshaker, the mixture was incubated at 650 rpm for 60 min and then reacted with 100 µL of 3% (*m*/*v*) BSA to block non-specific binding sites of AuNPs for another 30 min. The Ab−AuNPs conjugate was separated from the excess anti−F17A and BSA by centrifugation at 14,000 rpm and 4 °C for 10 min using a ScanSpeed 1730 MR centrifuge (Labogene, Allerød, Denmark). The supernatant was removed, and the conjugate was re-suspended in 500 µL Tween-20 solution 0.40% (*v*/*v*), and this solution was centrifuged at 14,000 rpm and 4 °C for 10 min again. Finally, the obtained conjugate was re-suspended in 500 μL of PBS and stored at 4 °C for further use.

### 4.5. Electrochemical Biosensor

To perform electrochemical measurements, Metrohm PGSTAT M204 potentiostat fitted with FRA32 impedance (Metrohm-Autolab, Utrecht, The Netherlands) and controlled by Nova^®^ software (v2.1.3), screen-printed carbon electrodes (DRP−110, DropSens, Llanera, Spain), and a specific cable connector (ref. DRP-CAC, DropSens) acting as interface between the SPCE and the potentiostat, were used. Firstly, the Ab − AuNPs was electrochemically deposited onto SPCE by cycling the electrode potential between 0 and +1 V vs Ag/AgCl at scan rate of 0.05 V/s for 20 min [55,56]. Then, 10 μL of isolated F17–positive *E. coli*, *E. coli* (F17–negative) or purified F17A protein samples were dropped onto Ab−AuNPs/SPCE surface and incubated for 30 min. After washed with PBS buffer successively, 10 μL HRP-Ab was casted onto the modified working electrode area and incubated for 15 min, followed by washing with PBS. Finally, electrochemical measurements were carried out by casting 50 µL of PBS buffer (pH 7.4) containing 1.0 mM HQ (freshly prepared before each measurement) and 10 mM H_2_O_2_.

DPV was performed in the potential range of −0.60 V to 0.40 V with a pulse amplitude of 50 mV and a pulse width of 0.05 s. An applied potential of −0.20 V vs. Ag/AgCl pseudo-reference electrode was needed to register amperometric responses. The amperometric signals given through the manuscript (ΔI) matched the difference between the currents measured in the absence and in the presence of the target analyte. Electrochemical impedance spectroscopy measurements were carried out in PBS solution containing 5 mM K_3_Fe(CN)_6_/K_4_Fe(CN)_6_ (1:1) with a frequency ranging from 100 kHz to 0.1 Hz at an applied potential of 0.20 V vs. Ag/AgCl, and with an amplitude modulation of 0.20 V. Randles equivalent circuit [R_s_(C[R_ct_W])]; where, R_s_: the solution resistance; C: the capacitance; R_ct_: the charge-transfer resistance; and W: the Warburg impedance; was used to fit Nyquist plots.

### 4.6. Fluorescence-Based Biosensor

The fluorescence−based sandwiched immune-complex was generated by first incubation of aliquots of 50 μL Ab-AuNPs and 50 µL of isolated F17–positive *E. coli*, F17–negative *E. coli* or purified F17A protein for 30 min. After centrifugation at 2000 rpm and 4 °C for 10 min, the supernatant was discarded and the conjugated complex of target/Ab-AuNPs were washed 3 times with 100 μL of PBS to remove unbound substances. Then, the conjugations were re-dispersed with 80 μL PBS and 20 μL of HRP-Ab and incubated for 15 min. After another centrifugation and washing step at the same conditions, the sandwich complexes between Ab-AuNPs and HRP-Ab were redispersed in 250 µL of PBS solution, and 100 μL of o-PDA solution (25 mM) and 50 μL of H_2_O_2_ solution (25 mM) were added and incubated for 15 min. The enzymatic catalytic reaction was blocked by adding 50 μL of KCN (10 mM). Fluorescence assay was accomplished using a fluorescence spectrofluorotometer (Shimadzu RF-6000, Kyoto, Japan) provided with a 150 W Xenon discharge lamp with the excitation wavelength at 450 nm and the emission spectra at 550 nm.

## 5. Conclusions

The rapid detection of pathogens is very important for a rapid decision making on the therapeutic protocols to recommend, especially for diseases with multiple etiology (bacteria, viruses, parasites …). Diarrhea is among these diseases, it causes high mortality and is very contagious and can be caused by several bacterial (*E. coli*, *Salmonella* …) or viral agents (*rotavirus*, *coronavirus* …). *E. coli* remains the most frequent bacterial agent with a very high genetic variability characterized by several virulence factors. Here we have described a simple and rapid method for the detection of *E. coli* strains expressing the F17 fimbria responsible for the attachment of bacteria to intestinal epithelial cells. A sandwich biosensor using anti−F17A/AuNPs conjugate as the capture probe and HRP-labeled anti−F17A antibody as the signal amplification probe was developed. The biosensor was able to detect F17–positive strain with a lower limit of detection of 37 CFU·mL^−1^.

## Figures and Tables

**Figure 1 ijms-23-06028-f001:**
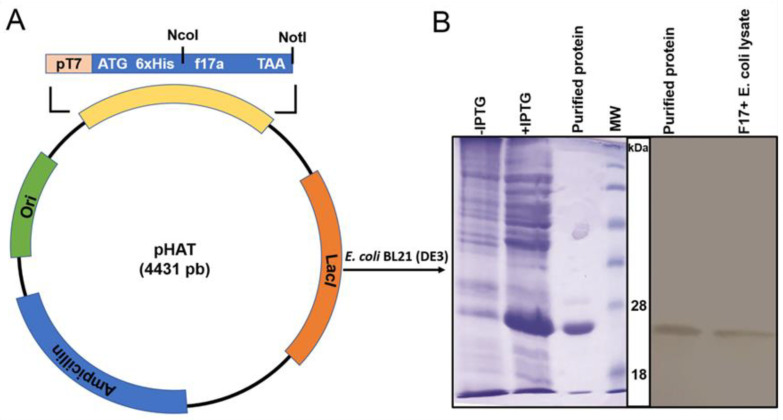
(**A**) Schematic representation of the plasmid construct used to express F17A protein. The f17a gene was cloned downstream 6xHis tag for periplasmic expression. (**B**) F17A recombinant protein expression after induction of BL21 (DE3) cells with IPTG was detected by SDS-PAGE. Cell lysates before induction with IPTG (−IPTG) or after induction with IPTG (+IPTG) showing an intense band at about 20 kDa, the expected size of F17A and the FPLC purified and desalted F17A protein. MW: See Blue Protein standard (Thermo scientific, Waltham, MA, USA), 18 and 28 kDa bands are indicated. Detection of F17A protein by western blot with the developed polyclonal antibody from purified recombinant F17A protein and the native F17A protein from an F17–positive *E. coli* diarrheic strain.

**Figure 2 ijms-23-06028-f002:**
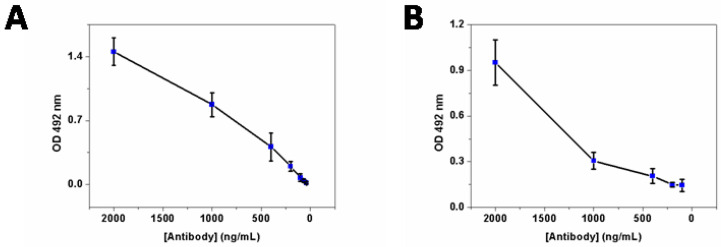
(**A**) Detection of the purified recombinant F17A by ELISA with the developed polyclonal antibody, decreasing concentration of the FPLC purified polyclonal antibody was used to detect the purified F17A protein-coated ELISA plate wells (100 µL of 5 µg·mL^−1^ solution in PBS). Detection limit was about 100 ng·mL^−1^ of purified rabbit antibody. (**B**) Detection of the native F17A at the surface of F17 fimbriae expressing *E. coli* strain. Each plate well was coated with 10^6^ cells and polyclonal antibody was used to detect native F17A on the cell surface. Detection limit is about 200 ng·mL^−1^.

**Figure 3 ijms-23-06028-f003:**
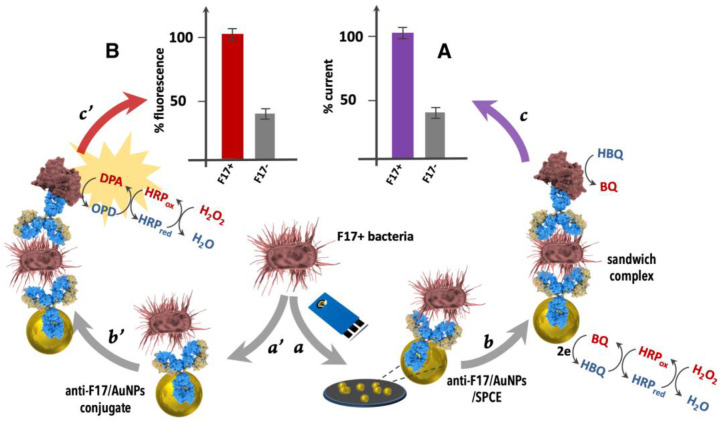
Principle of the (**A**) Electrochemical and (**B**) Fluorescence bioassays for *E. coli* F17A fimbrial protein detection. Steps: (a): adding F17−positive *E. coli* to anti−F17/AuNPs/SPCE; (b): adding anti−F17−HRP conjugate and (c): adding H_2_O_2_/HBQ and electrochemical readout at −0.1 V and (a’): adding F17–positive E. coli to anti−F17/AuNPs conjugate; (b’): adding anti−F17-HRP conjugate and (c’): adding H_2_O_2_/OPD and fluorescence readout at 550 nm.

**Figure 4 ijms-23-06028-f004:**
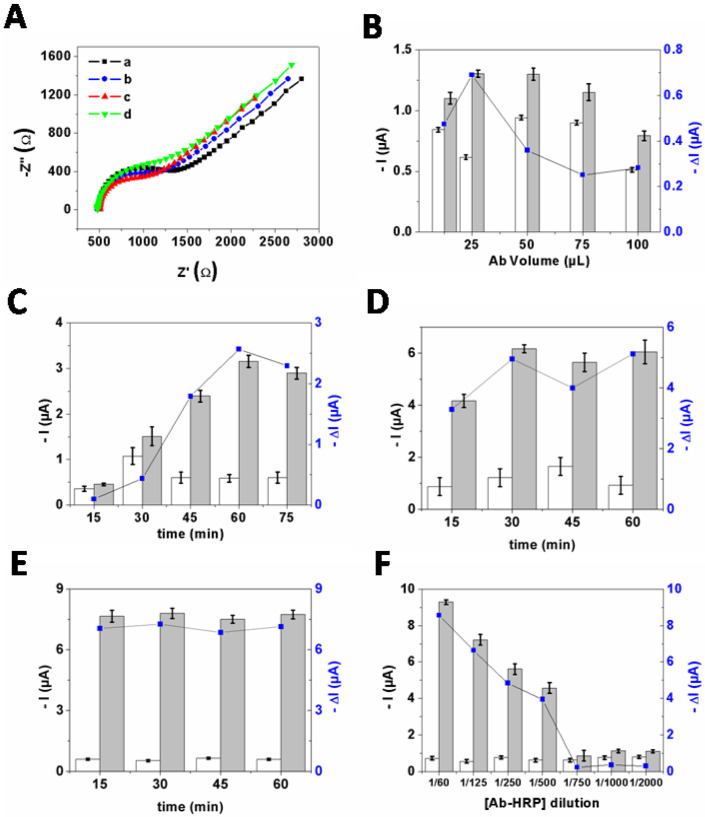
(**A**) Nyquist plots in 0.01 M PBS (pH 7.4) containing 5 mM K_3_Fe(CN)_6_/K_4_Fe(CN)_6_ (1:1) of: (a) bare SPCE; (b) Ab−AuNPs/SPCE; (c) F17A/Ab−AuNPs/SPCE; and (d) HRP−Ab/F17A/Ab− AuNPs/SPCE. Optimization of the experimental conditions: (**B**) the Ab (0.5mg·mL^−1^) volume used to react with AuNPs solution (500 µL, pH 9.5) for 30 min at RT; (**C**) the time of immobilization of Ab (0.5 mg·mL^−1^) onto AuNPs (500 µL, pH 9.5); (**D**) the time of immobilization of *E. coli* F17–positive (10^6^ CFU·mL^−1^) onto Ab−AuNPs/SPCE; (**E**) the time of immobilization of Ab−HRP (1/125) onto *E. coli* (F17–positive)/Ab−AuNPs/SPCE; and (**F**) the Ab−HRP concentration. Current measured before (white bars) and after (grey bars) immobilization of *E. coli* F17–positive (10^6^ CFU·mL^−1^). DI is the difference between the current measured before and after reaction with *E. coli*. The error bars stand for standard deviation recorded with three independent measurements (*n* = 3).

**Figure 5 ijms-23-06028-f005:**
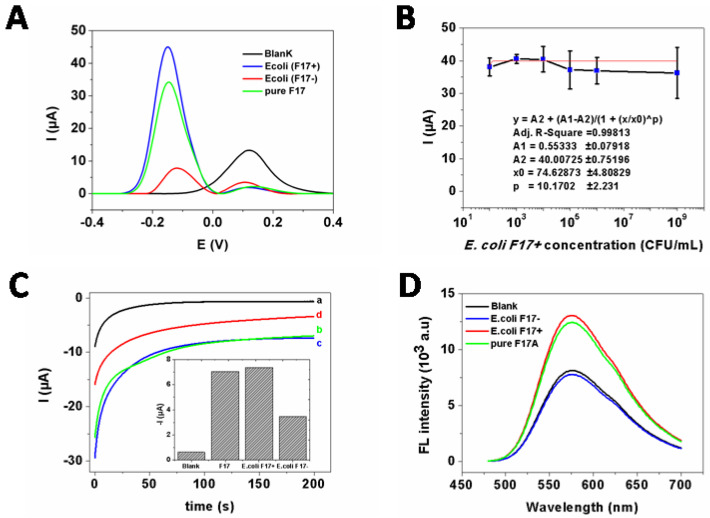
(**A**) DPV responses of the sandwich electrochemical immunosensor in the absence (a) and presence of *E. coli* F17−positive (10^6^ CFU·mL^−1^) (b); *E. coli* F17−negative (10^6^ CFU·mL^−1^) (c) and pure F17A protein (1.5 mg·mL^−1^) (d). (**B**) The calibration plot of *E. coli* F17–positive DPV detection where the data was fitted using 4-PL regression analysis. Error bars, SD, *n* = 3. (**C**) Amperometric responses of the sandwich electrochemical immunosensor: (a) in the absence of any analyte (blank) and in the presence of (b) pure F17A protein (1.5 mg·mL^−1^); (c) *E. coli* F17−positive (10^6^ CFU·mL^−1^); and (d) *E. coli* F17−negative (10^6^ CFU·mL^−1^). (**D**) Fluorescence emission spectra (λ_ex_/(λ_em_: 450/550 nm) of the sandwich immunosensor for the blank (black), in the presence of: 10^6^ CFU·mL^−1^ of *E. coli* F17−negative (blue); 10^6^ CFU·mL^−1^ *E. coli* F17−positive (ref) and pure F17A protein (1.5 mg·mL^−1^).

**Table 1 ijms-23-06028-t001:** Comparison of the prepared sensor performances with other electrochemical sensors, reported in literature, used for pathogenic *E. coli* detection.

Analytical Method	Target	LOD (CFU·mL−1)	Linear Range (CFU·mL−1)	Ref.
DPV	*E. coli* O157:H7	32	1 × 102–1 × 107	[26]
EIS	*E. coli* O157:H7	48	1 × 100–1 × 103	[39]
EIS	*E. coli* O157:H7	290	1 × 102–1 × 106	[41]
CV	*E. coli* O157:H7	2840	8.9 × 103–8.9 × 109	[42]
CV	*E. coli* O157:H7	450	4 × 103–4 × 108	[43]
EIS	*E. coli* DH5α	43	5 × 101–1 × 104	[44]
DPV	*E. coli* O111	112	2 × 102–2 × 106	[45]
DPV	*E. coli* F17	37	1 × 102–1 × 109	This work

Au/AuNPs/PrG/Ab: Au electrode modified with AuNPs, thiolated protein G and (PrG-thiol) and anti−*E. coli* IgG (Ab).

## Data Availability

The data presented in this study are available on request from the corresponding author.

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
