# Peer review of "Sandwich-Based Immunosensor for Dual-Mode Detection of Pathogenic F17–Positive Escherichia coli Strains"

_ijms, 2022, doi:10.3390/ijms23116028_

Round 1

Reviewer 1 Report

In this work, the authors have engineered a sandwich based immune sensor that can detect can E. coli strains that express the F17 pili. The pili have been shown to play an important role in pathogenicity across different bacterial species including E. coli. The pili helps in attachment of the pathogen to the host cell prior to the infection, thereby playing a key role in pathogenicity. For example, the uropathogenic strain of E. coli expresses P-pili encoded by the pap-operon that were shown to play an important role in uropathogenesis. In this work, the authors have attached the gold nanoparticles to the antiF17 rabbit polyclonal antibody that can specifically recognize the F-17A protein, thereby forming the Au-NP-F17 protein complex that is recognized by a second set of antibodies which is tagged with HRP for detection of this complex. The authors have tested the antibody against purified recombinant F17 proteins as well as with cell lysate. Overall, the work is well conducted and will be of importance to people studying pathogenic bacteria.

Minor comment

However, I’ve query about the cloning of the gene that encodes the F17A.

The authors have made a recombinant protein by tagging the histidine tag at N- terminus. However, the resultant recombinant protein was not present in solubilization fraction, rather it was localized in the inclusion body. The authors have purified it from the inclusion body under denaturating condition using urea followed by renaturation using dialysis. Did the authors try to tag it in the C-terminus and see if it improves the solubility?

Author Response

We gratefully thank the reviewers for the time they devoted to read and comment on our manuscript. We favorably replied to their comments, and we hope that the manuscript will be acceptable in the revised format.

Sincerely yours,

Noureddine Raouafi

Reviewer #1

The authors have made a recombinant protein by tagging the histidine tag at N- terminus. However, the resultant recombinant protein was not present in solubilization fraction, rather it was localized in the inclusion body. The authors have purified it from the inclusion body under denaturating condition using urea followed by renaturation using dialysis. Did the authors try to tag it in the C-terminus and see if it improves the solubility?

Reply

We thank the reviewer #1 for insightful comment. Indeed, tagging the protein c-terminally with His-Tag needs a signal peptide to direct expression to the periplasm, this may improve solubility but will reduce the amount produced.

Another major problem intrinsic to the F17A protein is that it is the monomer of the F17 fimbria and it tends to self-aggregate which complicates its expression in soluble state.

Reviewer 2 Report

The present manuscript entitled "Sandwich−based immunosensor for dual-mode detection of pathogenic F17-positive Escherichia coli strains" by Imed Salhi, Rabti Amal, Dhehibi Asma, and Raouafi Noureddine (ijms-1722708) describes the development of a novel sandwich immunosensor for the detection of recombinant and native F17A protein in F17 fimbriae-positive E. coli bacteria fabricated by conjugating the anti-F17A antibody with gold nanoparticles and horseradish peroxidase (HRP). This approach made electrochemical and fluorescent detection possible of purified and native F17A protein and differentiation between F17 fimbriae positive and negative E. coli strains.

The present article is written correctly and has a good structure. The article is interesting from a diagnostic and medical point of view; therefore, it should interest the reader. The paper meets the International Journal of Molecular Sciences' requirements, and I recommend the article for publication in the International Journal of Molecular Sciences following the common editing stage. My current decision is a minor revision. More specific comments and observations are presented below.

  1. Abstract. Value of linear range can be added.
  2. Keywords. Electrochemical and fluorescent detection should be mentioned.
  3. Introduction. One paragraph on sandwich-based immunosensors for similar uses may be added.
  4. Please check that all abbreviations are explained before using them.
  5. In the sensor description, in addition to specifying the LOD and the linear range, it would be good also to include the LOQ, the slope, intercept, and R2.
  6. Figure 4. Legend in Fig. 4 (A) and the data presented are very dimly visible. Individual drawings can be enlarged, and it would also be good to improve the axis description.
  7. Page 5, equation 2. “red” should be instead of “ref”.
  8. Figure 8. The main bands can be assigned numerically in the figure.
  9. RSD expressed as a percentage is the coefficient of variation (CV).
  10. Has the interference been tested? What can be done in the event of strong interference effects? How would you deal with them? What types of interference effects could occur?
  11. Discussion. Currently, the literature references do not follow in ascending order. The references should be renumbered, or Table 1 should be put in a different place.
  12. The article is missing the Conclusions section.
  13. Does the developed immunosensor have disadvantages?
  14. Page 8, line 279. Figure 3C or 5C?
  15. Please add the countries of origin, not just the companies, in the experimental part. What were the parameters of deionized water?
  16. In a few cases, it is necessary to correct the indexes in the chemical formulas.
  17. Page 9, line 342. Typo: minuntilcolor.
  18. What was the size of the Au NPs? Has it been checked?
  19. Page 10 is blank.
  20. References. Please check with the journal's requirements. Sometimes the journal abbreviations are not used. The Journal name is missing in [36].

I hope that the comments presented will help improve the article.

Author Response

We gratefully thank the reviewers for the time they devoted to read and comment on our manuscript. We favorably replied to their comments, and we hope that the manuscript will be acceptable in the revised format.

Sincerely yours,

Noureddine Raouafi

Reviewer #2

  1. Abstract. Value of linear range can be added.

Reply

We thank the reviewer for his comment:

We added the value of the linear range to the abstract.

  1. Keywords. Electrochemical and fluorescent detection should be mentioned.

Reply

We thank the reviewer for his comment:

We added the two keywords suggested by the second reviewer.

  1. Introduction. One paragraph on sandwich-based immunosensors for similar uses may be added.

Reply

We thank the reviewer for his comment. We added the following paragraph sandwich-based immunosensors describing the biosensors for bacteria detection. The corresponding references were added to the reference list.

“Detecting the whole bacteria can be advantageous since it allows to spike the DNA extraction and PCR amplification which are time consuming steps. Several recent works reported the use of sandwich-based approach for the detection of pathogenic bacteria. For instance, Bu et al. used a methylene blue loaded compositive as a signal-amplifying agent and magainin I as a bioreceptor to detect E. coli O157:H7 bacterial strain with a LOD of 32 CFU mL−1. An interesting sandwich-based concept for the detecting Shiga toxin-producing E. coli (STEC) was recently reported by Quintela and Wu. Indeed, the authors used two bacteriophages, the first was functionalized with biotin for surface tethering and served to capture the bacterium and the second one was labelled with HRP for signal transduction. The biosensor is not selective for a particular strain, but rather able to detect several with a limit of detection varying from 10 to 100 CFU mL−1 for STEC O157, O26, and O179 strains. A set of two aptamers doe S. aureus was selected and used to design an electrochemical biosensor the target bacterium. The limit of detection was 39 CFU mL−1 for spiked buffer solution and tenfold higher in tap water. An hybrid sandwich using one antibody and one aptamer was used to selectively detect Vibrio parahaemolyticus. The aptamer serves a starting point for rolling circle amplification in order to amplify the readout signal, to achieve a LOD as low as 2 CFU mL−1.“

  1. Please check that all abbreviations are explained before using them.

Reply

We thank the reviewer for his comment. We added abbreviations for all terms used in the manuscript.

  1. In the sensor description, in addition to specifying the LOD and the linear range, it would be good also to include the LOQ, the slope, intercept, and R2.

Reply

We thank the reviewer for his comment. We added the value of LOQto the text. R2 was already given. The values for the non−linear 4−parameter logistic (4−PL) model are displayed in Figure 4B.

  1. Figure 4. Legend in Fig. 4 (A) and the data presented are very dimly visible. Individual drawings can be enlarged, and it would also be good to improve the axis description.

Reply

We apologize for the quality of the figures signaled by the reviewer. We replaced Figure 4 by a better readable one.

  1. Page 5, equation 2. “red” should be instead of “ref”.

Reply

We apologize for this mistype error. We corrected it.

  1. Figure 8. The main bands can be assigned numerically in the figure.

Reply

We corrected Figure 5 (meant by the reviewer) as recommended.

  1. RSD expressed as a percentage is the coefficient of variation (CV).

Reply

The relative standard deviation (RSD) is expressed in percent and is obtained by multiplying the standard deviation by 100 and dividing this product by the average which means it is obtained by multiplying the coefficient of variation by 100.

  1. Has the interference been tested? What can be done in the event of strong interference effects? How would you deal with them? What types of interference effects could occur?

Reply

The interference has been tested. The goal of this work is to distinguish pathogenic bacteria (F17+) from the other. So, we did test the biosensor with two strains the first one is F17-negative and the second one is F17-positive. We also used the purified F17 protein to confirm that the response only come from the presence of F17 fimbrial protein expressed by F17-positive E. coli.

  1. Discussion. Currently, the literature references do not follow in ascending order. The references should be renumbered, or Table 1 should be put in a different place.

Reply

We corrected the bibliographic references to present them in chronological order of appearance.

  1. The article is missing the Conclusions section.

Reply

We thank the reviewer for his comment. We added the following paragraph as conclusion.

« The rapid detection of pathogens is very important for a rapid decision making on the therapeutic protocols to recommend, especially for diseases with multiple etiology (bacteria, viruses, parasites ...). Diarrhea is among these diseases, it causes high mortality and is very contagious and can be caused by several bacterial (E. coli, Salmonella ...) or viral agents (rotavirus, coronavirus ...). E. coli remains the most frequent bacterial agent with a very high genetic variability characterized by several virulence factors. Here we have described a simple and rapid method for the detection of E. coli strains expressing the F17 fimbria responsible for the attachment of bacteria to intestinal epithelial cells. A sandwich biosensor using anti-F17A/AuNPs conjugate as the capture probe and HRP-labeled anti-F17A antibody as the signal amplification probe was developed. The biosensor was able to detect F17 positive strain with a lower limit of detection of 37 CFU/mL.»

  1. Does the developed immunosensor have disadvantages?

Reply

We thank the reviewer for his comment.

The polyclonal antibody is a mix of F17A specific antibodies and other antibodies that can interfere with other proteins from other bacteria. We are developing a camel antibody fragment VHH (monoclonal) with higher specificity.

  1. Page 8, line 279. Figure 3C or 5C?

Reply

We apologize for the mistaken signaled by the reviewer. We corrected it.

  1. Please add the countries of origin, not just the companies, in the experimental part. What were the parameters of deionized water?

Reply

We added the country of origin for the companies which their materials and goods were used in this study.

  1. In a few cases, it is necessary to correct the indexes in the chemical formulas.

Reply

We apologize for the mistake. We corrected it.

  1. Page 9, line 342. Typo: minuntilcolor.

Reply

We apologize for the typo mistake signaled by the reviewer. We corrected it.

  1. What was the size of the Au NPs? Has it been checked?

Reply

We used the well-established Turkevich’s method to prepare the nanoparticles. The method yields nanoparticles of average diameter of 20 nm, that have a Plasmon band at 520 nm in UV-visible spectroscopy. We did this technique check the size.

  1. Page 10 is blank.

Reply

We apologize for the mistaken signaled by the reviewer. We corrected it.

  1. References. Please check with the journal's requirements. Sometimes the journal abbreviations are not used. The Journal nameismissing in [36].

Reply

We apologize for the mistaken signaled by the reviewer. We used Endnote for bibliography, some journals, or their abbreviations, can be missing.